# UW-Net: An Inception-Attention Network for Underwater Image Classification

## Abstract

The classification of images taken in special imaging environments except air is the first challenge in extending the applications of deep learning. We report on an UW-Net (Underwater Network), a new convolutional neural network (CNN) based network for underwater image classification. In this model, we simulate the visual correlation of background attention with image understanding for special environments, such as fog and underwater by constructing an inception-attention (I-A) module. The experimental results demonstrate that the proposed UW-Net achieves an accuracy of 99.3% on underwater image classification, which is significantly better than other image classification networks, such as AlexNet, InceptionV3, ResNet and Se-ResNet. Moreover, we demonstrate the proposed I-A module can be used to boost the performance of the existing object recognition networks. By substituting the inception module with the I-A module, the Inception-ResnetV2 network achieves a 10.7% top1 error rate and a 0% top5 error rate on the subset of ILSVRC-2012, which further illustrates the function of the background attention in the image classifications.

## 1 Introduction

Underwater images and videos contain a lot of valuable information for many underwater scientific researches (Klausner & Azimi-Sadjadi, 2019; Peng et al., 2018). However, the image analysis systems and classification algorithms designed for natural images (Redmon & Farhadi, 2018; He et al., 2017) cannot be directly applied to the underwater images due to the complex distortions existed in underwater images (e.g., low contrast, blurring, non-uniform brightness, non-uniform color casting and noises) and there is, to the best of our knowledge, no model for underwater image classification. Except for the inevitable distortions exhibited in underwater images, there are other three key problems for the classification of underwater images: (1) the background in underwater images taken in different environments are various; (2) the salient objects such as ruins, fish, diver exist not only in underwater environment, but also in air. The features extracted from the salient objects cannot be relied on primarily in the classification of underwater images; and (3) since the classification of underwater images is only a dualistic classification task, the structure of the designed network should be simple to avoid the over-fitting.

Increasing the depth and width of a CNN can usually improve the performance of the model, but is more prone to cause over-fitting when the training dataset is limited, and needs more computational resource (LeCun et al., 2015; Srivastava et al., 2014). To remit this issue, (Szegedy et al., 2015) proposed the inception module, which simultaneously performs the multi-scale convolution and pooling on a level of CNN to output multi-scale features. In addition, the attention mechanism (Chikkerur et al., 2010; Borji & Itti, 2012) is proposed and applied in the recent deep models which takes the advantage that human vision pays attention to different parts of the image depending on the recognition tasks (Mnih et al., 2014; Zhu et al., 2018; Ba et al., 2014). Although these strategies play an important role in advancing the field of image classifications, we find that the large-scale features such as the background area play a more important role in the visual attention mechanism when people understanding of underwater images, which is unlike the attention mechanism applied in natural scene image classification (Xiao et al., 2015; Fu et al., 2017).

In this paper, we propose an underwater image classification network, called UW-Net. The overview network structure is shown in Fig. 1. Unlike other models, the UW-Net pays more attention to the

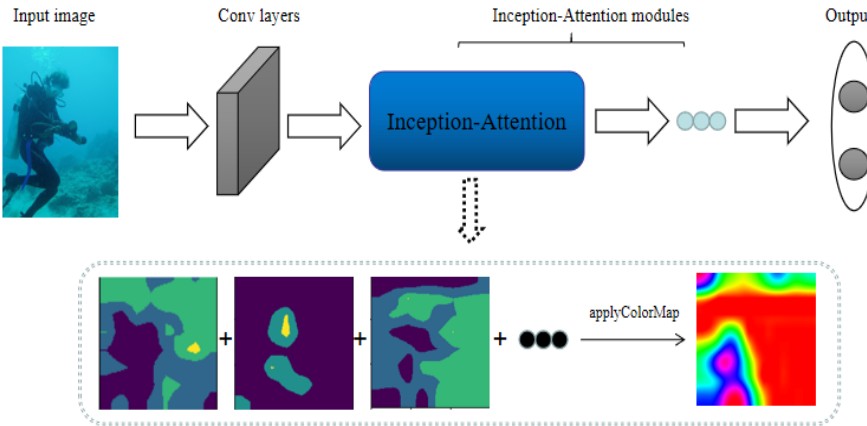

Figure 1: The structure of the UW-Net. The bottom part is the output of the eighth layer in the I-A module. The red area represents a higher response of features for the underwater image classification. As shown, our I-A module concerns more about the background regions of underwater images.

background features of images by construct the inception-attention (I-A) modules and thus achieves better performance. The contributions of this paper are as follows: (i) to the best of our knowledge, it is the first CNN-based model for underwater image classification; (ii) an inception-attention module is proposed, which joints the multi-dimensional inception module with the attention module to realize the multiple weighting of the output of various scales of features; (iii) this work is a first attempt to simulate the visual correlation between understanding images and background areas through I-A modules.

The rest of the paper is organized as follows: Section 2 introduces the related work. The proposed UW-Net is described in Section 3. Section 4 illustrates the experimental results and analysis, and we summarize this paper in Section 5.

## 2 RELATED WORK

As mentioned before, there is less work focusing on underwater image classification. Thus, we mainly introduce the classification models designed for natural scenes and the recent attention mechanism which is incorporated in our network in this section.

### 2.1 IMAGE CLASSIFICATION MODELS

Since Krizhevsky et al. (2012) won the ImageNet Large Scale Visual Recognition Competition (ILSVRC) (Deng et al., 2009), CNNs become more and more popular in the application of image recognition tasks. Many CNNs pursue better performance by means of superimposing more convolution layers, but the number of the parameters is concomitantly increased. As the depth increases (Simonyan & Zisserman, 2014), the gradient of the network will disappear or explode in the training process. Particularly, Szegedy et al. (2015) proposed the "GoogleNet" model, in which the inception architecture is first proposed. The $1 \times 1$ convolution is also applied as a dimension reduction technique (Lin et al., 2013) and to extract more nonlinear characteristics of the features. By fusing the features extracted from multi-scale convolutions, better image representation is obtained in the deep layers of the network. Based on the initial inception structure, multiple network structures such as Inception V3 (Szegedy et al., 2016) and Inception V4 (Szegedy et al., 2017) are further proposed. He et al. (2016) proposed the Resnet, in which the residual modules are proved to be effective in solving the gradient disappearance problem of deep convolution networks.

An important feature of human vision is that people usually focus more on a certain area of the whole scene while ignoring other areas (Mnih et al., 2014; Treisman & Gelade, 1980), which is called

the attention mechanism (AM). Hu et al. (2018) proposed the SeNet by constructing the attention mechanism of feature channels, and won the championship of ILSVRC2017. Wang et al. (2017) proposed the residual attention network (RAN), which also combined the attention mechanism with the residual modules, and achieves 4.8% top-5 error rate on ILSVRC 2012 dataset. These works are all evidences of the effectiveness of the attention mechanism in deep learning models on one hand, and on the other, none of these studies on attention mechanism focus on background features.

## 2.2 UNDERWATER IMAGING

Although the performance of the existing image classification models has exceeded that of human beings in some classification tasks (Huang et al., 2017; He et al., 2016), most of the existing classification models assume that images have legible texture and uniform features. However, when light propagating through the water, the absorption and scattering determined by the internal optical property (IOP) of the water affect the process of underwater imaging. Not only the water body, but also the dissolved organic matter and small floating particles (called sea snow), whose concentration and species vary greatly, also affect the underwater image quality (Kjerstad, 2014; Johnsen et al., 2009). With the depth increases in water, the color of light disappears according to their wavelengths. Artificial lighting often results in uneven lighting, creates bright spots in the image, and makes the scattering of suspended matter worse (Yang & Sowmya, 2015). These challenges make the design of an effective underwater image classification algorithm difficult.

Moreover, human recognition of underwater images is often based on the background features of the images. In the next section, we use the inception module with multiple sizes of receptive fields to extract features and simulate the human attention mechanism by combining an attention module emphasizing the background features of underwater images.

## 3 PROPOSED APPROACH

The network structure of UW-Net is constructed based on multiple I-A modules, as shown in Fig. 2. The network can be extended with more I-A modules for other complex visual classification tasks. An underwater image is first forwarded to a convolutional layer with $7 \times 7$ size of kernels to obtain large reception fields. An auxiliary classification branch is added after two I-A modules, and the output of the auxiliary branch will be used to optimize the network as part of the loss function. As follows, we will introduce the key components in the UW-Net, i.e., I-A module including inception module and attention module, and classification branch in details.

### 3.1 INCEPTION-ATTENTION MODULE

The classical inception models (Szegedy et al., 2015; 2016; 2017) are constructed by multiple sizes of convolutional kernels. The feature maps are processed by different convolution kernels in one inception module, and then merged and forwarded to the next inception module directly. However, not all the extracted features of every convolution kernel are positively related to the current image classification task. For example, the features extracted by a convolution kernel with large size tend to describe the global information, which has little effect on a fine-grained image classification task even though they can be transferred to the deeper layers, and will result in a certain degree of waste of computational resource.

#### 3.1.1 INCEPTION MODULE

Underwater images exhibit the characteristics of large intra-class differences. Furthermore, the positions and proportions of the background areas vary in underwater images, and the recognition of underwater image is based more on the global features of the image. In addition, the quality of most underwater images is poor due to the effects of lighting absorption and scattering. In view of these characteristics of the underwater images, we adopt convolution kernels with larger size and the average pooling to reduce the impact of local features of the image on the final classification (Boureau et al., 2010). In the experiment, we find that the best classification results can be obtained by using convolution kernel sizes of $1 \times 1$, $5 \times 5$ and $7 \times 7$ in the inception module.

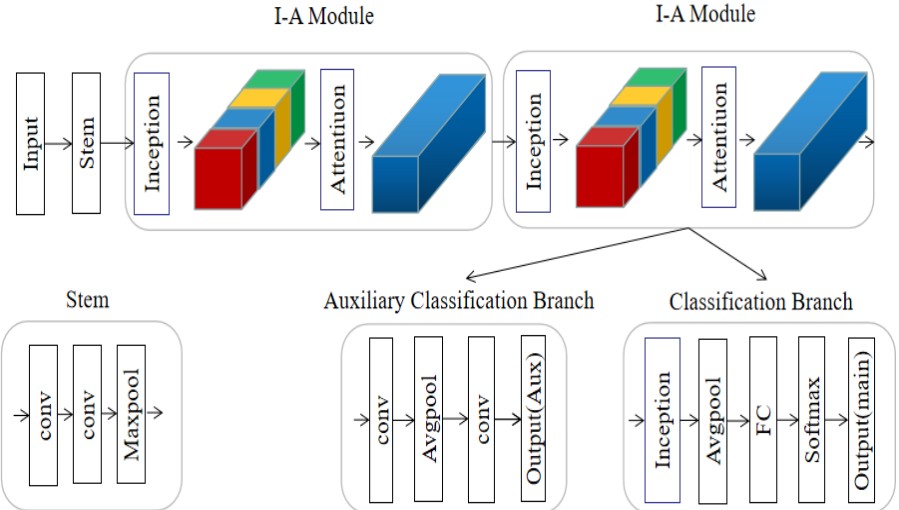

Figure 2: The structure of the UW-Net. After two I-A modules, one branch serves as an auxiliary classification branch and one branch serves as the backbone of the network. The final model has two outputs, in which the output of the auxiliary branch is used to compute the loss function.

### 3.1.2 ATTENTION MODULE

We construct the attention module based on soft attention (Xu et al., 2015), as shown in Fig. 3, which consists of a trunk branch and a mask branch to simulate the recognition of underwater image by humans beings (Wang et al., 2017; Hu et al., 2018). Inspired by the residual network (He et al., 2016), the trunk branch takes the output of the previous layer as input directly so that the basic features of the image can be transmitted to the deep layers of the network, and the gradient disappearance and the gradient explosion can be remitted. On the other hand, the down-sampling operation is firstly performed in the mask branch, and then up-sampling by bilinear interpolation is used at the last step to keep the same size of the feature map with the input. The activation function of the first and second convolutional layers are Relu (Nair & Hinton, 2010) and Sigmoid (Han & Moraga, 1995), respectively. The adaptive weight $N(x)$ for a point $x$ of the original features map $P(x)$, in the range of [0,1], can be learned after the mask branch. The output of the attention module $F(x)$ can be expressed as:

$$F(x) = (1 + N(x)) \times p(x).$$ 
(1)

For a $N(x)$ approximating 1, $F(x)$ will be near twice the value of the original feature $P(x)$, which means that for the features that are valid for the current classification, more attention will be given. On the contrary, for a $N(x)$ approximating 0, the output of the attention module will approximate the original feature $P(x)$.

### 3.2 AUXILIARY CLASSIFICATION BRANCH AND LOSS FUNCTION

In the proposed UW-Net, an auxiliary classification branch is introduced to the output of the second I-A module to reduce the risk of over-fitting. The convergence curve of the UW-Net is shown in Fig. 4. It is obvious that adding auxiliary classifier can not only accelerate the convergence, but also improve the accuracy on the test set.

The loss function $J$ of the UW-Net can be expressed as follows:

$$J = J_0 + J_1 + \alpha \times L,$$
(2)

where $J_0$ is the cross entropy of the final output of the model and the real label of the image, $J_1$ is the cross entropy of the output of the model's auxiliary classification branch and the actual label, $\alpha$ is the weight attenuation coefficient of the network, and $L$ is the L2 regularization term.

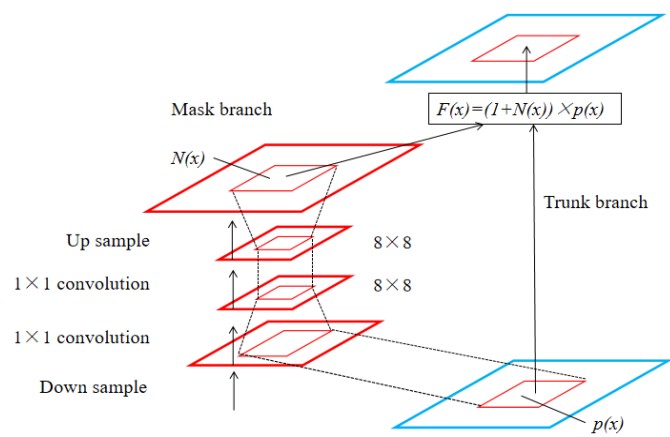

Figure 3: The architecture of the attention module.

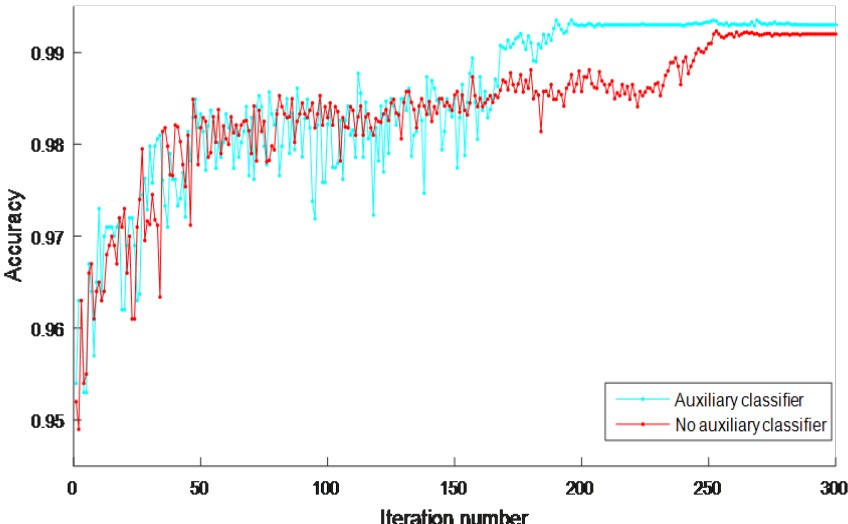

Figure 4: The accuracy curves of models on the test set.

### 3.3 THE UW-NET NETWORK

The UW-Net is constructed by two I-A modules proposed above, and the detailed blocks and parameter settings are reported in Table 1. Before the first I-A module, the size of the image is reduced to $36 \times 36$, and the channel of the input image is increased to 32 by two convolutional layers and one pooling operation. After the I-A module, the size of the feature map is reduced and the channel of the feature map is increased. The final prediction of the model is obtained after an average pooling (Avgpool) and a fully connected (FC) layer.

## 4 EXPERIMENTS

In this section, a series of experiments are conducted to demonstrate the performance of the UW-Net in underwater image classification. Additionally, the effectiveness of the proposed I-A module in improving the performance of existing inception based networks is investigated. We compare the UW-Net with several typical image classification models and inception model based networks. The

Table 1: The network structure of UW-Net.

| Type | Patch size/stride or remarks | Input size |
|---|---|---|
| Conv | $7 \times 7/2$ | $299 \times 299 \times 3$ |
| Conv | $3 \times 3/1$ | $147 \times 147 \times 16$ |
| Pool | $3 \times 3/2$ | $73 \times 73 \times 32$ |
| $1\times$I-A module | As in figure 2 | $36 \times 36 \times 32$ |
| $1\times$I-A module | As in figure 2 | $36 \times 36 \times 128$ |
| $1\times$Inception | As in Section 3.1.1 | $17 \times 17 \times 512$ |
| Avgpool | $8 \times 8/1$ | $8 \times 8 \times 1024$ |
| Dropout | 0.5 | $1 \times 1 \times 1024$ |
| FC | logits | $1 \times 1 \times 512$ |
| Softmaxl | classifier | $1 \times 1 \times 2$ |

models for comparison are re-trained in the same dataset without changing the structure, only the parameters are optimized.

### 4.1 BENCHMARK DATASET

To ensure the diversity of underwater images, we collect more than 4,000 underwater images from ImageNet dataset [1], JAMSTEC dataset [2], underwater rock image dataset [3] and online underwater images. These images are labeled as underwater images. In addition, more than 5,000 non-underwater images from the ImageNet including birds, cars, food, airplanes, cats, etc. are selected and labeled as non-underwater images.

### 4.2 TRAINING AND ANALYSIS

The proportion of the samples used in training and testing is 70% and 30%, respectively. To reduce the risk of over-fitting, we augment the data by random clipping and flipping. We use the initialization method proposed by He et al. (2015) to initialize weights and train the UW-Net by using the SGD (Stochastic Gradient Descent) (Ketkar, 2014) with a mini-batch size of 32. The weight decay, momentum and initial learning rate are set to 0.001, 0.9 and 0.001, respectively. The learning rate is decreased ten times of its original value at 1k and 2k iterations. The training end at 3k iterations. The loss curve is shown in Fig. 5.

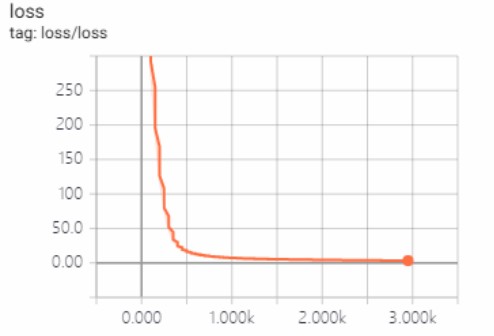

Figure 5: The curve of the classification training loss of the UW-Net.

---

[1][Online]. Available: http://www.image-net.org/.
[2][Online]. Available: http://www.fishdb.co.uk/.
[3][Online]. Available: https://github.com/kskin/WaterGAN/.

Table 2: The comparisons with the competing models on underwater image classification.

| Network | Mult-Adds(Million) | Parameters(Million) | Test Accuracy |
|---|---|---|---|
| AlexNet | 720 | 60 | 96.5% |
| VGG16 | 15300 | 138 | 97.0% |
| Googlenet | 1550 | 6.8 | 98.1% |
| ResNet-50 | 3860 | 25.6 | 97.8% |
| SE-ResNet-50 | 3870 | 28 | 98.0% |
| Our model | **750** | **6.6** | **99.3%** |

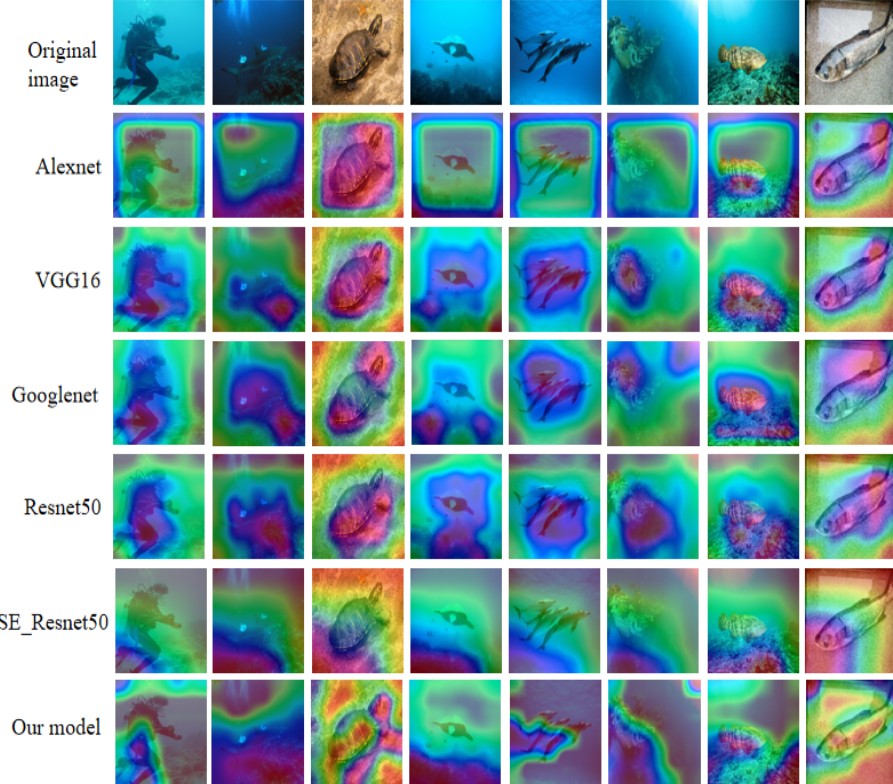

Figure 6: Visualized examples of regional contributions to the final classification. From up to down are: the output of the fifth layer in Alexnet, the output of the thirteenth layer in VGG16, the output of the twelfth layer in Googlenet, the output of the seventeenth layer in Resnet-50, the output of the seventeenth layer in SE-Resnet-50 and the output of the eighth layer in the UW-Net.

### 4.3 EXPERIMENTAL RESULTS

The UW-Net achieves 100% and 99.3% accuracy on the training and testing datasets, respectively. We also report the comparisons with the AlexNet, VGG16, InceptionV3, Resnet-50 and SE-Resnet-50 on the testing dataset in Table 2, and class activation maps (CAMs) (Selvaraju et al., 2017) produced by these models for underwater image classification are shown in Fig. 6. The darker red color in the CAMs represent more importance of the regions to the final classification. As shown in Fig. 6, the interesting areas of the UW-Net locate more in the background areas compared to the competing models. The data in Table 2 shows that the UW-Net is superior to the competing models for the task of underwater image classification by fewer computation units and parameters, and higher accuracy. By adopting the I-A model in the UW-Net, we achieve higher accuracy with a smaller depth of network. At the same time, the UW-Net requires lower computation than the other inception based models.

Table 3: The performance of the I-A modules in other networks.

| Network | Top-1 err. | | Top-5 err. | |
|---|---|---|---|---|
| | Inception | I-A | Inception | I-A |
| GoogleNet | 18.3% | 17.8% | 1.2% | 1.0% |
| InceptionV3 | 13.1% | 12.6% | 0.0% | 0.0% |
| InceptionV4 | 11.2% | 11.0% | 0.0% | 0.0% |
| Inception-ResnetV2 | 11.0% | 10.7% | 0.0% | 0.0% |

## 4.4 THE PERFORMANCE OF THE I-A MODULE

The I-A module designed in this work can not only be applied in the UW-Net, but also to other common image classification networks. To further verify the generalization ability of the I-A module in boosting the performance of related models, we embed the proposed I-A module with the max-pooling in the down-sampling of the mask branch into several image classification models including GoogleNet, InceptionV3, InceptionV4 and Inception-ResnetV2. One hundred categories of images are selected from the ILSVRC-2012 dataset, including ships, sharks, dogs, cocks, etc. Each category contains about 1300 images, and a total number of 130,000 images is included. The training and testing sets consist of 125,000, and 5000 images respectively. The size of the images is $299 \times 299$. All the networks are tested and tuned on this dataset in the same way. The experimental results are shown in Table 3. It can be seen that the error rates of Top-1 obtained by substituting the inception module with the I-A module are significantly decreased. Such a result indicates the generalization and effectiveness of the proposed I-A module, also gives another evidence that attention weighted large-scale image features simulate better the visual understanding mechanism in image classifications.

## 5 CONCLUSION

A new underwater image classification network UW-Net is proposed in this work, wherein an inception-attention module is constructed. In this model, we simulate the visual correlation between understanding images and background areas through I-A modules, which joint the multidimensional inception module with the attention module to realize the multiple weighting of the output of various scales of features. The 100% accuracy on the training set and 99.3% accuracy on the testing set of the UW-Net is achieved benefiting from the refinement of the usefulness of multiscale features by the I-A module. In the future, we will try to improve the performance of other underwater image visual analysis models by introducing the proposed I-A module.

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
