# OpenReview forum: "UW-NET: AN INCEPTION-ATTENTION NETWORK FOR UNDERWATER IMAGE CLASSIFICATION"
_ICLR.cc/2020/Conference — Reject_

### Official Review · AnonReviewer1 · 2019-10-21
**Official Blind Review #1**

**Rating:** 3

**Review:**

This paper shows a method to classify between underwater and non-underwater images (binary). For this, they proposed a new module in a CNN network, called 'Inception-Attention' module. It combines the inception module (multiple sizes of kernels) with an attention module (learn jointly some masks in order to focus the network to part(s) on the image). Also, their network proposes 2 classifications, by separating the network in 2 branches at the end - with the goal of less overfitting. Their method was compared against other state-of-the-art networks, usually designed for more complex tasks (multiclass classification). Lastly, they showed how their new 'IA module' could be useful in other tasks/networks, by replacing the standard inception network with theirs.

Overall, I found the paper confusing: while some technical aspects are interesting, I still don't get the motivation of this new network for this task. Moreover, I did not understand the goal of the paper until the middle of the paper: I thought it was to do a multi-class classification from underwater images, while it is just a binary classification of underwater/ non-underwater images. I still think there are some valuable arguments, and the last part, 4.4, showing the usefulness of the method for more complex classifications, is to me the most interesting part. I could change my grade if a better motivation or a reorienting of the paper was made.

Remarks/ questions:
- The writing should be improved as it is even hard to understand some sentences. Some particular help will be given below.
- My main concern is: have you tested easier classifications methods? It seems that classifying underwater vs. not underwater images would be easy. In fact, even the mean color should be identifiable... do you have a simple baseline to compare to? You are saying that one of the main problems is that salient objects are less visible underwater, there is blurring, ect. : these are for me all arguments why it would be easy to detect underwater images, because standard convolutions will behave differently.
- Have you tested a simpler CNN? You are right in saying that the state-of-the art methods for multi-class classification are too large for this task. So why to you want to complexify it, and why not use a simpler network?
- There is no related work on background classification (or 'context' classification); but I am sure that there might be works on this. It would be more interesting than the general image classification models.
- in 3.1 and 3.2, you are saying 2 opposite things. a) 'the features extracted by a conv. kernel with large size tend to describe the global information [..] and will result in a certain degree of waste of computational resource' and b) 'we adopt convolution kernels and the average pooling to reduce the impact of local features of the image'. Do you want or not want large kernels?
- 'Moreover, human recognition of underwater images is often based on the background..'. Why? Are you sure? I think it is more based on the color of the image, the texture, ect.
- Auxiliary classification branch: If you have overfitting problems; why don't you use standard methods for treating overfitting (the first one is to have a smaller and simpler network..)?
-I don't see a real improvement between the auxiliary classification branch and no auxiliary classifier in the Figure 4. Such a small difference, compared to the large oscillations, is not enough. Why don't you use different runs of your model with different initializations, in order to take the mean?
- 3.3: not clear: 'the channel of the input image is increased to 32'. I think you want to talk about the number of channels, and not 'the channel'. Same error few lines below.
- How did you select the 5000 non-underwater images? Randomly? Or just outside views?
- 4.2: I don't see a validation set. It is important to have 3 sets, one training, one validation, one test. All the tuning of parameters/architectures must be done on the validation, with the test kept hidden until the publication.
- 4.3: results on the training set should not even be shown...if you want, you can show the validation error, but not the training. An accuracy of 99.3% means that it is an easy task. Yes, it's true that your method works, but i) since you are not using a validation set, you could have just tuned your model until you have a good accuracy on the test set; ii) it might be better to prove the usefulness of your method on a more complex task.
- 4.4 : this is the important part of the paper I think, you should develop it :)!

**Experience Assessment:**

I have read many papers in this area.

**Review Assessment: Checking Correctness Of Derivations And Theory:**

I assessed the sensibility of the derivations and theory.

**Review Assessment: Checking Correctness Of Experiments:**

I carefully checked the experiments.

**Review Assessment: Thoroughness In Paper Reading:**

I read the paper at least twice and used my best judgement in assessing the paper.

---

### Official Review · AnonReviewer2 · 2019-10-23
**Official Blind Review #2**

**Rating:** 1

**Review:**

- This works presents yet another incremental modification of well-known architectures (i.e., inception network). Particularly, authors propose to add an attention mechanism in the inception module to pay more attention to a given set of features. To evaluate the proposed network, authors resort to the task of underwater image classification.
- The technical contribution of this work is rather limited, being the main contribution the application of deep classification models to underwater image classification.
- Furthermore, authors mention that this work is a first attempt to simulate the visual correlation between understanding images and background areas through I-A modules. Nevertheless, this was never shown, beyond some classification activation maps in Figure 6. If I understood correctly, the classification task basically reduces to predict whether an image is taken underwater or not. Focusing on the background, however, may introduce errors, since I believe that pictures containing mainly sky may trigger the same activations. It would be more interesting to see also the classification activation maps for the negative class (non-underwater images).
- Additionally, looking at Fig 6, GoogleNet and ResNet seem to provide more meaningful regions than the proposed network in some cases. Again, showing results on non-underwater images would help to better understand how the proposed method works.
- Authors mislead some messages: ‘classification algorithms designed for natural images cannot be directly applied to the underwater images due to the complex distortions existed in underwater images’. Later in the manuscript, they show that those standard classification algorithms achieve almost the same performance as the propose method (1-2% of difference). Does it mean that this 1-2% of improvement makes these algorithms applicable on this task?? In another example: ‘The classification of images taken in special imaging environments except air is the first challenge in extending the applications of deep learning.’
- Some results on Table 3 are useless (e.g., those 5-top error equal to 0%).
- Authors split the dataset into training and evaluation. Nevertheless, they should also use a validation set to stop the training and pick the best model (based on the validation images) to generate the predictions on the testing set. Otherwise, they may be overfitting the model on the training set.
- Overall, this paper presents an incremental contribution with respect to existing networks, just to improve 1% the classification performance on an easy task (baseline performance around 98%). Thus, I do not feel that this work may attract the interest of the ICLR attendees.

**Experience Assessment:**

I have read many papers in this area.

**Review Assessment: Checking Correctness Of Derivations And Theory:**

I carefully checked the derivations and theory.

**Review Assessment: Checking Correctness Of Experiments:**

I carefully checked the experiments.

**Review Assessment: Thoroughness In Paper Reading:**

I read the paper thoroughly.

---

### Official Review · AnonReviewer3 · 2019-10-23
**Official Blind Review #3**

**Rating:** 3

**Review:**

This paper proposed an underwater image classification network. The current manuscript missed some very important information (see below). Besides, the experimental results are also weak.
The paper mentioned "complex distortions existed in underwater images (e.g., low contrast, blurring, non-uniform brightness, non-uniform color casting and noises)" many times. But when the paper introduced the UW-Net structure, it does not explain how the network over-comes these difficulties. The UW-Net structure only considers the factors of the background and attention. Thus I think the proposed network structure is not convincing.
For the experimental part, I am afraid the results are also weak. For example, please notice that many network structures have proposed to improve the classification. I think authors should compare more existing works to demonstrate the superiority of the proposed one.


**Experience Assessment:**

I have published in this field for several years.

**Review Assessment: Checking Correctness Of Derivations And Theory:**

N/A

**Review Assessment: Checking Correctness Of Experiments:**

I carefully checked the experiments.

**Review Assessment: Thoroughness In Paper Reading:**

I read the paper at least twice and used my best judgement in assessing the paper.

---

### Decision · Program_Chairs · 2019-12-19

**Decision:**

Reject

**Comment:**

The reviewers have issues with the lack of enough experimental results as well as with novelty of the solution proposed. I recommend rejection.